# Observing and braiding topological Majorana modes on programmable quantum simulators

Nikhil Harle [1,2], Oles Shtanko [3] & Ramis Movassagh [2,4]

Electrons are indivisible elementary particles, yet paradoxically a collection of them can act as a fraction of a single electron, exhibiting exotic and useful properties. One such collective excitation, known as a topological Majorana mode, is naturally stable against perturbations, such as unwanted local noise, and can thereby robustly store quantum information. As such, Majorana modes serve as the basic primitive of topological quantum computing, providing resilience to errors. However, their demonstration on quantum hardware has remained elusive. Here, we demonstrate a verifiable identification and braiding of topological Majorana modes using a superconducting quantum processor as a quantum simulator. By simulating fermions on a one-dimensional lattice subject to a periodic drive, we confirm the existence of Majorana modes localized at the edges, and distinguish them from other trivial modes. To simulate a basic logical operation of topological quantum computing known as braiding, we propose a non-adiabatic technique, whose implementation reveals correct braiding statistics in our experiments. This work could further be used to study topological models of matter using circuit-based simulations, and shows that long-sought quantum phenomena can be realized by anyone in cloud-run quantum simulations, whereby accelerating fundamental discoveries in quantum science and technology.

It is a unique time in the history of science and engineering when we are witnessing significant advances in the development of fully controllable, coherent many-body quantum systems that contain dozens to hundreds of qubits[1]. Quantum simulators hold the promise of exponentially outperforming classical computers, which would bring about a host of applications beyond the reach of classical computers. Perhaps the most promising application of these systems is the simulation of quantum many-body systems[2], which includes topological phases of matter[3,4]. In addition to their exotic nature, topological quantum states are a promising route to fault-tolerant quantum computation that is based on non-Abelian excitations such as Majorana fermions[5]. Majorana fermions are exotic particles: each is its own antiparticle, unlike an electron being distinct from its antiparticle

(positron). Despite the remarkable progress, the original proposal for the realization of Majorana-based quantum memories on solid-state devices[6–8] ultimately encountered difficulties due to disorder and lack of control, as well as the inability to separate Majorana modes from other trivial zero-energy states[9–14]. At the same time, quantum simulators may help in this search with their unprecedented levels of parameter control for a range of topological models[15–17].

Realization of topological phases hosting Majorana modes in bosonic multi-qubit devices was first envisioned few decades ago[18], with subsequent theoretical developments[19,20]. Since then signatures of topological modes were detected in photonic experiments[21–23] and programmable digital quantum information processors[24–30]. While these devices are limited to non-equilibrium settings they still are able

[1]Department of Physics, Yale University, New Haven, CT 06520, USA. [2]IBM Quantum, MIT-IBM Watson AI lab, Cambridge, MA 02142, USA. [3]IBM Quantum, IBM Research – Almaden, San Jose, CA 95120, USA. [4]Google Quantum AI, Venice Beach, CA 90291, USA. ✉e-mail: movassagh@google.com

to exhibit long-lived signatures of topological modes[31,32]. Some of these signatures were analyzed in programmable processors with methods usually tailored to free-fermionic models[33–35]. However, the qualitative study of the properties of these topological excitations remained a challenge. Braiding of the Majorana fermions is yet another motivation as it provides the exchange statistics of the topological excitations and is a necessary step for topological quantum computation. While there has been progress in manipulation of toy Majorana modes in photonics[36–38] and superconducting architectures[39–42], they were limited to a few qubit systems, not a real topological phase. Thus, direct probing of the topological modes and their manipulation remained an open problem.

Using existing noisy quantum hardware, we aim to perform quantitative simulations of topological quantum matter. We recreate the state of one-dimensional topological superconductor widely known for hosting a pair of exotic "half-electron" Majorana modes at its boundaries. We show how to use Fourier transformation of multi-qubit observables to reliably determine the structure of Majorana modes. We also demonstrate how the detection of two-point correlation functions make it possible to distinguish between trivial and topological modes. Finally, we introduce and implement the fast approximate swap (FAS): a general non-adiabatic method to approximately braid Majorana fermions in one dimension. Unlike conventional adiabatic methods, it allows implementation on the current generation of noisy quantum hardware.

## Results

### Floquet engineering

Time-periodic (Floquet) systems had proven to be particularly suitable for simulations on digital quantum processors. In particular, when system Hamiltonian alternates between two or more local Hamiltonians being sums of mutually commuting terms, this choice of quantum dynamics provides a remarkable resource utilization. In this way,

unlike trotterized continuous dynamics, a constant-time Floquet dynamics can be simulated using constant-depth circuit. While Floquet systems may be compared in their form to rough trotterization of continuous dynamics, they exhibit a wide variety of topological phases[43]. The Floquet topological phase may be quite robust despite the presence of disorder[44].

Our focus is on the time-periodic Hamiltonian

$$H(t) = \sum_{j=1}^{N-1} \left( J(t) X_j X_{j+1} + \lambda(t) Z_j Z_{j+1} \right) + h(t) \sum_{j=1}^{N} Z_j, \quad (1)$$

where $X_j$ and $Z_j$ are single-qubit Pauli operators, $\{J, \lambda, h\}(t+T) = \{J, \lambda, h\}(t)$ is a set of time-periodic parameters, $T$ is the time period, $N$ is the number of qubits.

We propose a protocol that divides a single driving period into three parts. For simplicity, we consider the driving period acting from $t = 0$ to $t = T$. During the first part, from the start of the period to time $\tau_1$, we set $h(t) = h$ and the other coefficients to zero, $J(t) = \lambda(t) = 0$. Next, for times in between $\tau_1$ and $\tau_2$, we set $J(t) = J$ and the rest of the coefficients to zero. Lastly, between $\tau_2$ and the end of the period $T$, we set the last term to be on, $\lambda(t) = \lambda$, and all other terms to zero. Therefore, only one term in the Hamiltonian in Eq. (1) is active at any given moment.

A quantum circuit can reproduce such a quantum dynamics protocol at discrete times $t_n = nT$. At such times, the system's state is described by the wavefunction $|\psi_n\rangle = U_F^n|\psi_0\rangle$, here $|\psi_0\rangle$ is the initial state and $U_F = \exp(-i \int_0^t H(t')dt')$ is the Floquet unitary,

$$U_F = \prod_{j=1}^{N-1} e^{-i\varphi Z_j Z_{j+1}} \prod_{j=1}^{N-1} e^{-i\theta X_j X_{j+1}} \prod_{j=1}^{N} e^{-i\phi Z_j}, \quad (2)$$

where the gate angles are $\phi = h\tau_1$, $\theta = J(\tau_2 - \tau_1)$, and $\varphi = \lambda(T - \tau_2)$. The corresponding experimental protocol that involves local single- and two-qubit gates is depicted in Fig. 1a, where each cycle corresponds to a single Floquet unitary.

The model has received considerable attention in the study of condensed matter systems due to its alternative description in terms of spinless fermions. By Jordan-Wigner transformation, the qubit Pauli operators can be transformed into non-local Majorana fermion operators $\gamma_\mu$ satisfying $\{\gamma_\mu, \gamma_\nu\} = 2\delta_{\mu\nu}$, where $\mu, \nu = 1, ..., 2N$[45]. It is not a unique mapping; here we use two equivalent Jordan-Wigner representations, denoted as $\gamma_\mu^{L,R}$ and associated with the right and left boundaries. In these representations a Majorana operator becomes a string of Pauli operators connected to one of the boundaries. As we show in Methods, the expectation values of these operators can be obtained from single-qubit measurements preceded by a series of two-qubit gates. We will not include the superscripts for the Majorana operators when the choice of representation is not important.

In the case that $\lambda = 0$, the Hamiltonian in Eq. (1) is non-interacting and takes the simple quadratic form $H(t) = \sum_{\mu,\nu=1}^{2N} h_{\mu\nu}(t)\gamma_\mu\gamma_\nu$, where $h_{\mu\nu}$ is an antisymmetric Hermitian matrix. Due to its free fermionic nature, dynamics generated by such a Hamiltonian are classically efficient to simulate (see Supplementary Note 1). In this regime, depending on the ratio between $J$ and $h$, the system exhibits various phases including the symmetry-protected topological phases[15,17], as summarized by the phase diagram shown in Fig. 1b. Among these four phases, there is one (shown in white) that is trivial and topologically equivalent to a product state. There are three more topological phases. The first phase is topologically equivalent to the static Kitaev chain (blue). Under open boundary conditions, this phase exhibits two symmetry-protected modes at zero quasi-energy called Majorana zero modes (MZM). The remaining topological phases only occur in time-driven systems. For example, the second phase (red) exhibits a pair of Majorana $\pi$ modes (MPM) occurring at quasi-energy $\pi$[16]. The third phase (green) is distinct from the rest and hosts both MZM and MPM. Majorana modes in non-interacting systems manifest

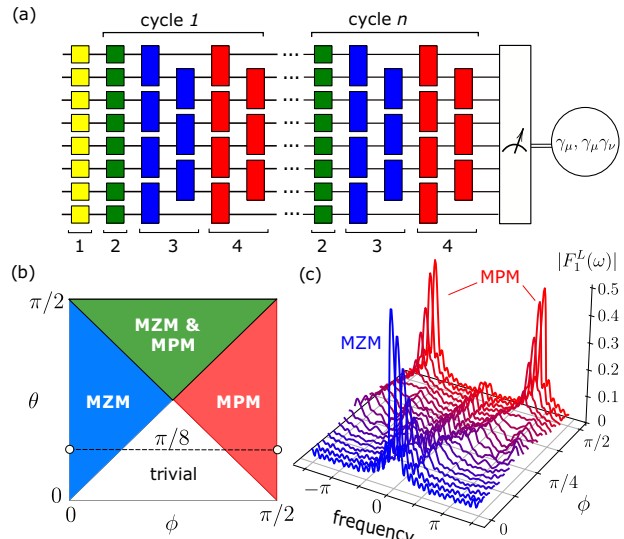

**Fig. 1 | Circuit and phase diagram. a** Schematics of an 8-qubit circuit including the initialization, evolution, and measurement parts. The initialization process is limited to the application of single-qubit Hadamard gates (yellow, #1). Evolution is composed of cycles consisting of $Z$ gates (green, #2), $XX$ gates (blue, #3), and $ZZ$ gates (red, #4). The measurements provide the expectation of the operators $\gamma_\mu^{L,R}$ or $\gamma_\mu\gamma_\nu$ (see "Methods" section). **b** Phase diagram for $\lambda = 0$, showing four possible phases, see text. **c** Experimentally measured Fourier component $|F_1^L(\omega)|$ as a function of $\phi$ for fixed $\theta = \pi/8$ using a 21-qubit system implemented on *ibm_hanoi*. The system exhibits transitions from MZM to trivial phase and from trivial to MPM phase. Detected peaks indicate the presence of Majorana modes at frequencies $\omega = 0$ and $\omega = \pi$.

themselves by the presence of a pair of conserved boundary-localized operators $\Gamma_s^\omega$ that satisfy $U_F^\dagger \Gamma_s^\omega U_F = e^{-i\omega}\Gamma_s^\omega$ [46,47] where index $s \in \{L, R\}$ defines right and left eigenmodes respectively, and $\omega \in \{0, \pi\}$. We will skip the frequency index $\omega$ when the context is clear.

In the interacting case $\varphi \neq 0$, Majorana mode operators are not conserved across the spectrum, i.e. $U_F^\dagger \Gamma_s^\omega U_F - e^{-i\omega}\Gamma_s^\omega = O(\tau^{-1})$. As a result, the observables associated with topological modes must decay with characteristic lifetime $\tau$. As was shown in ref. 32, if the bulk has vanishing dispersion, for small interaction angles $\varphi$ the lifetime diverges as $\tau \propto \mathcal{O}(\exp(c/\varphi))$, where the constant $c$ depends on the details of interaction. In practice, the lifetime may exceed dozens of Floquet cycles even if the bulk has finite dispersion and interactions are not too strong. This approximate conservation of Majorana modes leads to the persistent signal for some local observables when the rest reach infinite-temperature values. The primary goal of this work is to use this long-lived signal to restore the structure of the modes from the experiment. In this case we look for Majorana modes of the form $\Gamma_s = \sum_{\mu=1}^{2N} \psi_\mu^s \gamma_\mu$, where $\psi_\mu^s$ are real-valued wavefunctions. We also develop a method to distinguish trivial and topological modes.

Finally, we illustrate the exchange of Majorana modes and verify that the exchange results in the desired change of phase of the wavefunction. Conventionally, such an exchange is modeled by a slow adiabatic implementation of the unitary map $\mathcal{E}_{ex}(\cdot) = U_{ex}^\dagger(\cdot)U_{ex}$, where $U_{ex} = \exp(-\frac{\pi}{4}\Gamma_L\Gamma_R)$. Such a map provides $\mathcal{E}_{ex}(\Gamma_R) = \Gamma_L$ and $\mathcal{E}_{ex}(\Gamma_L) = -\Gamma_R$. While it is possible to carry out this procedure for one-dimensional Floquet systems[48,49], it might require quantum circuits with depths beyond what is available on noisy devices. Below we show an alternative way to perform such an exchange on a noisy quantum hardware.

## Majorana wavefunctions

Our first objective is to detect the presence of Majorana modes and measure the details of their structure using Fourier transformation[30,31]. We assume that there are no other eigenmodes with zero or $\pi$ frequencies. In this case, we can use the asymptotic formula (see Supplementary Note 2)

$$\psi_\mu^L(\omega) = F_\mu^L(\omega)/\sqrt{F_1^L(\omega)}, \quad \psi_\mu^R(\omega) = F_\mu^R(\omega)/\sqrt{F_{2N}^R(\omega)}, \quad (3)$$

where $\omega \in \{0, \pi\}$ is the mode frequency, the positivity of $F_1^L(\omega)$ and $F_{2N}^R(\omega)$ is proven in Supplementary Note 2, and

$$F_\mu^s(\omega) = \lim_{N,D\to\infty} \frac{1}{D}\sum_{n=0}^{D-1} e^{i\omega n}\langle\psi_0|U_F^{\dagger n}\gamma_\mu^s U_F^n|\psi_0\rangle, \quad (4)$$

with $|\psi_0\rangle = \bigotimes_{i=1}^N |+\rangle_i$ being the product state of eigenstates of Pauli operator $X$ with eigenvalue one, and superscript $s \in \{L, R\}$ conforming with the representation of the Majorana operator. The order in the limit is important: one first takes the limit over the number of qubits $N$, and then the limit over the number of cycles $D$.

In spite of the fact that the true limit cannot be reached experimentally, we measure the quantities $F_\mu^s(\omega)$ approximately using the largest available $N$ and $D$. The values of $D$ must not exceed the Majorana mode lifetime $\tau$ such that $D/\tau \ll 1$. First, we initialize the qubits in the product state $|\psi_0\rangle$ and apply an $n$-cycle circuit as shown in Fig. 1a for $n = 0, ..., D-1$. For each circuit, we determine the expectation of $\gamma_\mu^{R,L}$. In the last step, we estimate the approximate value of $F_\mu(\omega)$ by summing up the results for each $n$-cycle circuit with corresponding Fourier coefficients $e^{i\omega n}/D$.

As an example, Fig. 1c shows the function $|F_1^L(\omega)|$ and its use in detecting Majorana modes and topological phases. The plots illustrate the dependence of this function on angle $\phi$ for the fixed $\theta = \pi/8$ and are similar to differential conductance spectra found in solid-state experiments[8]. The function is equal to the topological mode density at the boundary, $F_1^L(0) = (\psi_1^L)^2$. In particular, we observe a strong signal

for $\omega = 0$ in the topological phase for value $\phi = 0$, as it indicates the presence of the left MZM. Strong peaks also appear at frequencies $\omega = \pm\pi$ indicating the presence of MPM for $\phi = \pi/2$. The peaks' intensities decrease in the bulk for intermediate angles. For $\theta = \pi/8$ the boundary signal disappears at $\phi = \pi/8$ and $3\pi/8$ as the system transitions into the trivial phase.

Next, the values of $F_\mu^s(\omega)$ for $\mu > 1$ help us recover Majorana wavefunctions $\psi_\mu^s$. Plots in Fig. 2a–c illustrate the normalized absolute values of wavefunctions corresponding to MZMs and MPMs in both non-interacting ($\varphi = 0$) and interacting ($\varphi = \pi/16$) regimes. The results for the non-interacting regime are in a good agreement with the theoretical prediction. In the interacting regime, where we add an extra set of noisy two-qubit ZZ gates in each Floquet cycle, we expect to see a visibly higher level of noise in the resulting wavefunction as can be seen in Fig. 2c. More data to evaluate device performance is presented in the "Methods" section.

## Detecting trivial modes

Majorana modes may not be the only modes responsible for zero-frequency signals[9-14]. In this work, we demonstrate that quantum simulators can be used to distinguish unpaired Majorana zero modes from the other topologically trivial localized excitations. Topological Majorana $\pi$ modes can be treated similarly. We use a generalized notation $\Delta_k = \sum_\nu \psi_{k\nu}\gamma_\nu$ for both zero-frequency trivial and topological Majorana modes, $[\Delta_k, U_F] = 0$, and $\psi_{k\nu}$ are real wavefunctions that are localized at the boundaries. In contrast to Majorana modes residing at opposite boundaries, any pair of trivial modes must always be localized near the same position. Below we assume that the effect of disorder on the localization of the wavefunction is negligible.

We examine the two-point correlation function ($\omega = 0$)

$$T_{\mu,\nu} = \lim_{N,D\to\infty} \frac{1}{D}\sum_{n=0}^{D-1}\langle\tilde{\psi}_0|U_F^{\dagger n}\gamma_\mu\gamma_\nu U_F^n|\tilde{\psi}_0\rangle, \quad (5)$$

where $|\tilde{\psi}_0\rangle = |\psi_a\rangle|s_2\rangle|s_3\rangle...|s_{N-1}\rangle|\psi_a\rangle$, where $|s_i\rangle$ are random states in $Z$-basis with eigenvalues $s_i = \pm 1$, and $|\psi_a\rangle = \cos a|0\rangle + i\sin a|1\rangle$ for $a \in [0, \pi]$ being a phase. For simplicity, we consider the non-interacting case $\lambda = 0$. Then the value of the correlation function for $\mu = 1$ and $\nu = 2$ is (see Supplementary Note 3)

$$T_{1,2} = i\cos 2a \lim_{N\to\infty}\sum_{kk'}\left(\psi_{k1}^2\psi_{k'2}^2 - \psi_{k2}^2\psi_{k'1}^2\right). \quad (6)$$

If there is only one pair of topological modes separated by the system size, then $T_{1,2} = 0$. Indeed, in this case $\sum_{kk'}\psi_{k1}^2\psi_{k'2}^2 - \psi_{k2}^2\psi_{k'1}^2 = (\psi_1^R)^2(\psi_2^L)^2 - (\psi_2^R)^2(\psi_1^L)^2 \propto O(2^{-\Theta(N)})$. A pair of trivial localized states at the left boundary, however, would result in $T_{1,2} > 0$. At the same time, $T_{1,2N}$ is non-zero for both cases, while in the middle of the system, i.e. $T_{1,x} = 0$ for $x = 2cN$ and $1 > c > 0$. As a consequence, correlation function indicates the presence of zero-frequency modes but has a different structure for trivial and Majorana modes.

In order to illustrate this method, we consider two examples of non-interacting systems, $\lambda = 0$. In the first example, we use the Hamiltonian in the topological phase ($\theta = \pi/4$, $\phi = \pi/16$). We compare this case to a trivial system with a slightly modified Hamiltonian. In particular, we set to zero the $XX$-term and $Z$-term for the first and the last qubits, thus decoupling them from the rest of the system (see Supplementary Note 4). The state of the rest of the qubits is governed by the Floquet evolution in Eq. (2) with parameters ($\theta = \pi/16$, $\phi = \pi/4$). This modification mimics a possible error when some of the links between the qubits are dysfunctional. The modification produces two trivial full-electron modes at opposite boundaries, which is equivalent to four non-topological Majorana modes $\Delta_1 = \gamma_1$, $\Delta_2 = \gamma_2$, $\Delta_3 = \gamma_{2N-1}$, and $\Delta_4 = \gamma_{2N}$. Using only the observables in Eq. (4), it is difficult to distinguish between these

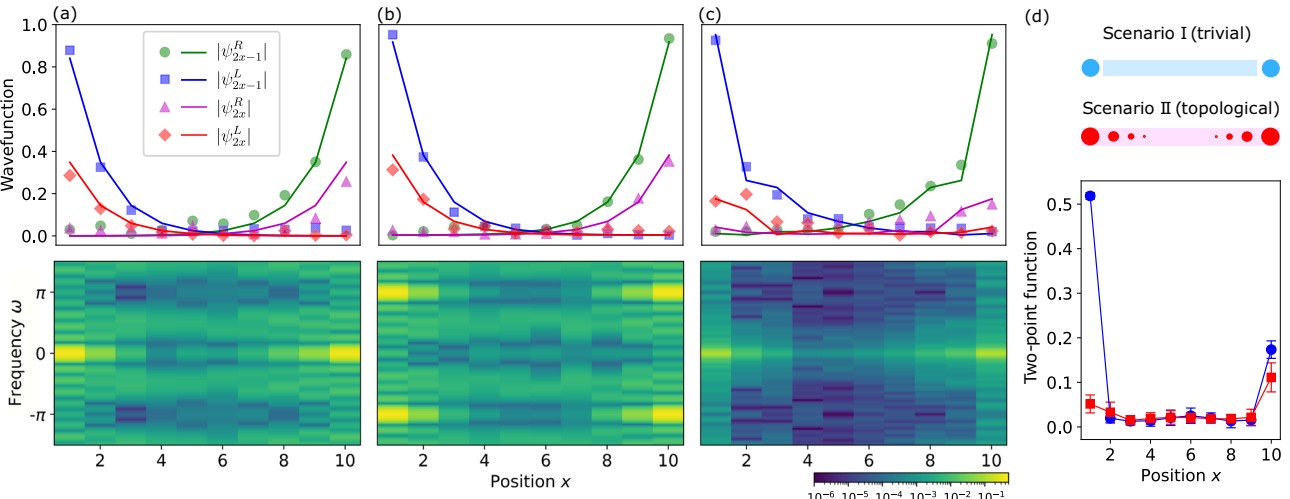

**Fig. 2 | Detection of Majorana modes.** Panels **a**–**c** show the absolute value of the experimentally observed Majorana mode wavefunctions $|\psi_\mu^s|$ (dots) in comparison with its theoretical prediction (lines) for $N = 10$ qubits. Wavefunctions are further normalized because under noise effects Eq. (3) is inexact. Bottom panel illustrates the density function $g(x, \omega) = |F_{2x-1}^L(\omega)|^2 + |F_{2x-1}^R(\omega)|^2$, the bright peaks show the frequency of the modes. **a** MZM extracted using *ibm_montreal* device in the topological phase $\theta = \pi/4$, $\phi = \pi/8$, and $\varphi = 0$, using $D = 11$ cycles. **b** MPM extracted using the same device in topological phase $\theta = \pi/4$, $\phi = 3\pi/8$, and $\varphi = 0$, using $D = 11$ cycles. **c** MZM wavefunction extracted using *ibm_mumbai* device for interacting topological phase $\theta = \pi/4$, $\phi = \pi/16$, and $\varphi = \pi/16$, using $D = 21$ cycles. **d** Difference between trivial phase $\theta = \pi/16$ and $\phi = \pi/4$ with two trivial boundary modes (blue, circles) and topological phase $\theta = \pi/4$ and $\phi = \pi/16$ (red, squares) quantified by $|T_{1,2x}|$ in Eq. (5), measured using *ibm_toronto* device. The result is calculated as the average of 10 random initial states and $D = 11$ cycles. The error bars are one standard deviation. The expectation values used to generate all figures are calculated by averaging over 8192 circuit runs.

modes and topological modes. However, if we measure the sequence $|T_{1,2x}|$ for $x = 1, \dots, N$ for a random configuration of the initial state, it shows an important difference. As shown in Fig. 2d, the curve for trivial case is characterized by two peaks at $x = 1$ and $x = N$, while topological system has only one peak around $x = N$. Thus, observation of a single peak provides reliable evidence distinguishing topological Majorana modes from the other possible trivial modes.

**Braiding Majorana modes**

Finally, we introduce a method for braiding the Majorana modes, which we call Fast Approximate Swap (FAS). Here we examine the parametrized map

$$\mathcal{E}_\alpha(\cdot) := \lim_{N,D\to\infty} \frac{1}{D}\sum_{n=0}^{D-1} U_{n\alpha}^\dagger(\cdot)\, U_{n\alpha}, \qquad (7)$$

where $U_{n\alpha} = U_F^n \exp(-\alpha\gamma_1\gamma_{2N})U_F^n$, and $\alpha \in [0, \pi]$ is a real parameter. This quantum channel is equivalent to selecting the unitary $U_{n\alpha}$ for $n = 0, \dots, D\text{-}1$ with uniform probability $1/D$.

Let us assume that the system is reflection-symmetric such that the localized modes satisfy $\psi_1^L = \psi_{2N}^R = \xi$ and $\xi^2 \geq 1/2$. Then, by setting the angle $\alpha_0 = \arcsin(1/\sqrt{2}\xi)$, the action of the map on topological Majorana operators is

$$\mathcal{E}_{\alpha_0}(\Gamma_R) = p\Gamma_L, \quad \mathcal{E}_{\alpha_0}(\Gamma_L) = -p\Gamma_R, \qquad (8)$$

where $p = \sqrt{2\xi^2 - 1} \leq 1$ (see Supplementary Note 4). This procedure constitutes approximate FAS method of braiding that aims to replace the conventional adiabatic process. This method applies in both the interacting and non-interacting regimes.

We also establish the effect of proposed braiding map on Majorana operators in absence of localization in non-interacting limit $\lambda = 0$,

$$\mathcal{E}_{\alpha_0}(\gamma_\mu) = p\left(\psi_\mu^R\Gamma_L - \psi_\mu^L\Gamma_R\right). \qquad (9)$$

This allows us to detect the relative phase of Majorana fermions after braiding. The braided mode wavefunction can be defined similarly to

Eq. (3) as

$$\tilde{\psi}_\mu^s = \frac{1}{\mathcal{N}}\langle\psi_0|\mathcal{E}_{\alpha_0}(\gamma_\mu^s)|\psi_0\rangle, \qquad (10)$$

where $\mathcal{N}$ is normalization coefficient. After braiding, assuming that the system is reflection-symmetric, we expect the braided wavefunctions to satisfy $\tilde{\psi}_\mu^L = \psi_\mu^R$ and $\tilde{\psi}_\mu^R = -\psi_\mu^L$. This behavior is illustrated in Fig. 3, where we compare wavefunctions in Eqs. (3) and (10).

Our braiding procedure depends on the parameter $\alpha_0$ that is generally unknown without prior access to the system. Although here we calculated it analytically, it may be difficult to find this angle theoretically for generic Hamiltonians, in which case it would be necessary to rely on experimental data. For instance, one can evaluate the angle using the measured Majorana wavefunction. In Supplementary Note 4, we discuss an alternative method of finding the proper value of $\alpha_0$.

## Discussion

In this work, we propose a framework for detecting, verifying, and braiding Majorana modes on near-term programmable quantum simulators by employing the Floquet dynamics. This scheme can be generalized to the continuous evolution of static Hamiltonians by replacing the discrete Fourier transformation in our work by its continuous version. It would have been possible to run our experiments on larger qubit devices. However, Majorana modes exist at the boundaries rather than the bulk, and our current experiments are sufficient to make conclusive statements about the detection and braiding of the topological Majorana modes.

The finite lifetime of the Majorana modes is attributed to natural tendency of Floquet systems to "heat up". Adding disorder such as randomization of phases in $Z$ gates, i.e. $\phi \to \phi + \delta_i$, where $\delta_i \in [-W, W]$, can reduce the heating because of the many-body localization (MBL) phenomenon[50]. However, such a simplistic scheme may require disorder values $W$ that can cause transition into the trivial phase. Avoiding phase transition would require finding good model parameters[51] or using more sophisticated techniques[32].

This work illustrates the power of synthetic near-term qubit-based quantum computers for demonstrating and studying topological

phases of electronic systems. Indeed, if we neglect noise, the observed dynamics of bosonic system can perfectly simulate fermionic topological phases if the measurements are made in the non-local qubit basis. Unlike solid-state devices, however, the Majorana modes in this work are subject to decoherence because noise breaks the parity symmetry protecting the topological phase in fermionic systems. This is a serious drawback for using then in topological quantum computation. Nonetheless, with improvements in coherence time, the role of noise can be sufficiently reduced as to make this type of quantum simulation useful in studying topological quantum matter. Solid-state systems such as nanowire devices[6,7] can be studied through continuous-time local Hamiltonian simulations. Floquet systems similar to those studied in this work, in the limit of large frequency, are equivalent to such simulations. A potential model of a nanowire could incorporate a qubit "ladder" representing the two spin values (up and down) and local gates that account for hopping, spin-orbital coupling, and density-density interactions.

This work can be extended to regimes beyond the current classical simulation capabilities. By using devices with higher connectivity, it is possible to study generic two-dimensional materials with a broader variety of topological phases and to explore new possibilities for topological quantum computation. Further, the method studied for extracting the Majorana modes may be extended to the study of local integrals of motion in many-body localized systems, similarly to the proposal in ref. 52.

## Methods

We replicate the Floquet dynamics in Eq. (2) on IBM Qiskit using the circuit in Fig. 1(a). We transpile the circuit on Qiskit using native gates and run it on the IBM quantum hardware (see Fig. 4). In particular, for the special value $\theta = \pi/4$ ($\varphi = \pi/4$), each $XX$-gate ($ZZ$-gate) requires a CNOT gate in combination with single qubit gates. For other non-zero angle values, two-qubit gates require two CNOTs in combination with other single-qubit gates. Therefore, to reduce the depth, part of the experiment is designed to investigate the case $\theta = \pi/4$.

For the simulation of quantum systems, Majorana operators must be encoded using qubits. We use the Jordan-Wigner transformation to implement this encoding. In particular, we define left representation as $\gamma^L_{2k-1} = \mathcal{Z}^L_k X_k$ and $\gamma^L_{2k} = \mathcal{Z}^L_k Y_k$, where $\mathcal{Z}^L_k = \prod_{i=1}^{k-1}(-Z_i)$ are $Z$-string operators and $k = 1, \ldots, N$. This representation is equivalent to the most common convention. Alternatively, the right representation is $\gamma^R_{2k-1} = \mathcal{Z}^R_k Y_k$ and $\gamma^R_{2k} = -\mathcal{Z}^R_k X_k$, where $\mathcal{Z}^R_k = \prod_{i=k+1}^{N}(-Z_i)$. A more traditional approach for accessing these operators experimentally is to measure each qubit inside the string in the basis ($X$, $Y$, or $Z$), returning $\pm 1$ values for each qubit. In this case, we can use the product of the obtained results as the measured value. The disadvantage of the conventional method is that it has lower precision due to accumulated measurement errors. Therefore, we adopt a scheme that involves only one measurement, as described below.

In order to measure $\gamma^L_\mu$, we use the expressions

$$U^\dagger_k X_1 U_k = (-1)^{k-1}\mathcal{Z}^L_k X_k, \qquad V^\dagger_k Y_1 V_k = (-1)^{k-1}\mathcal{Z}^L_k Y_k, \qquad (11)$$

where

$$U_k := U^{YX}_1 \ldots U^{YX}_{k-1}, \qquad V_k := U^{XY}_1 \ldots U^{XY}_{k-1}, \qquad (12)$$

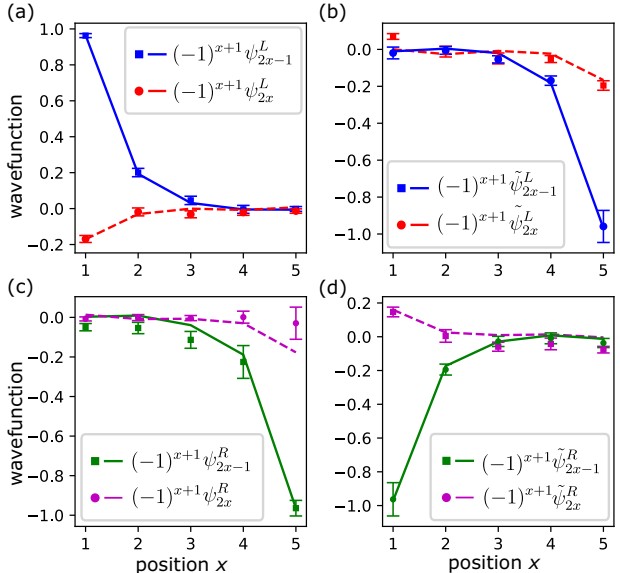

**Fig. 3 | Braiding.** Comparison of normalized original wavefunction in Eq. (3) for the left (**a**) and right (**c**) modes and braided wavefunction in Eq. (10) for the left (**b**) and right (**d**) modes with the theoretically estimated angle $\alpha_0 = 0.263127\pi$. We use the 5-qubit system on *ibm_hanoi* device with the parameters $\phi = \pi/16$, $\theta = \pi/4$, and $\varphi = 0$ and maximum number of cycles $D = 11$, averaged over 30 experiments each with 8192 shots. Error bars are one standard of deviation. Experimental data are represented by points, whereas theoretical predictions are represented by lines. Plots illustrate that modes acquire a relative minus sign after braiding $\tilde{\psi}^L_\mu = \psi^R_\mu$ and $\tilde{\psi}^R_\mu = -\psi^L_\mu$.

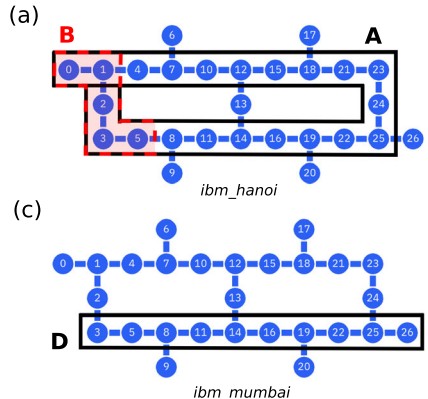

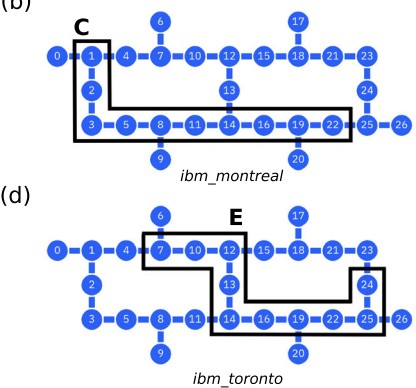

**Fig. 4 | Configuration of IBM hardware. a** Layout of the *ibm_hanoi* device. Sequence A of 21 qubits was used to generate the frequency-resolved boundary oscillations shown in Fig. 1c; sequence B of 10 qubits was used to perform the braiding experiment shown in Fig. 3. **b** Layout for the *ibm_montreal* device. Sequence C of 10 qubits was used to reproduce the Majorana mode tomography in

Fig. 2a, b. **c** Layout for the *ibm_mumbai* device. Sequence D of 10 qubits used to generate the Majorana mode tomography in Fig. 2c. **d** Layout for the *ibm_toronto* device. Sequence D of 10 qubits used to generate the topological/non-topological mode separation experiment in Fig. 2d.

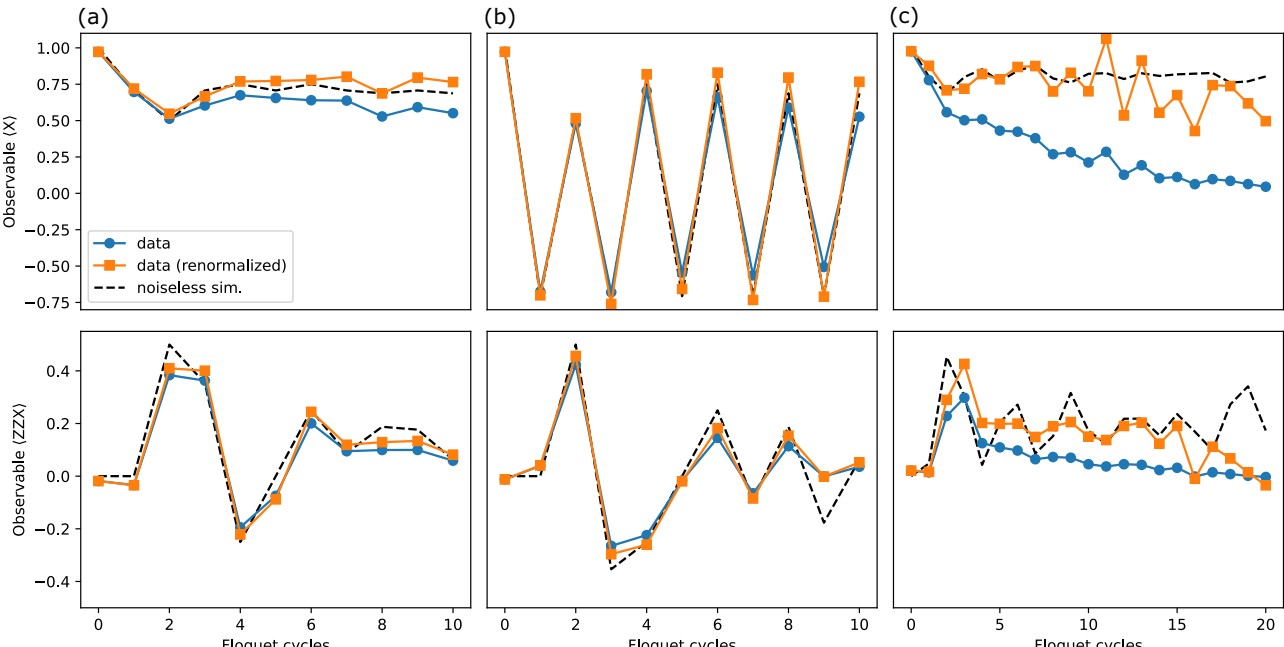

**Fig. 5 | Experimental data.** The figure shows the raw data for the expectations of the operators $X_0$ (top) and $Z_0Z_1X_2$ (bottom) for noiseless simulation (dashed black curve), the experimental data (circles, blue), and rescaled experimental data (squares, orange), where the rescaling takes the form $\exp(\Gamma n)$, where $n$ is the index of the Floquet cycle and $\Gamma$ is the compensated decay rate. This rescaling accounts for the decay of Majorana modes that does not contribute to the measured wavefunction (see Supplementary Note 2). Panel **a** shows the results for MZM in Fig. 2a (here, we use $\Gamma = 0.0328$ for rescaling), panel **b** shows the results for MPM in Fig. 2b (using $\Gamma = 0.0376$), and panel **c** shows the results for MZM in the presence of $ZZ$ gates shown in Fig. 2c (using $\Gamma = 0.120$).

and can be expressed as a product of two-qubit unitaries $U_j^{YX} := \exp(-i\frac{\pi}{4}Y_jX_{j+1})$ and $U_j^{XY} := \exp(i\frac{\pi}{4}X_jY_{j+1})$. According to these expressions, to measure the string operator $\gamma_{2k-1}^L = \mathcal{Z}_k^L X_k$, we apply the gates $U_j^{YX}$ consecutively and in the reverse order for $j = k-1, ..., 1$. Next, we measure the first qubit ($j = 1$) in X-basis. Similarly, to measure the operator $\gamma_{2k}^L = \mathcal{Z}_k^L Y_k$, we perform similar gate sequence but with unitaries $U_j^{XY}$ and measure the qubit $j = 1$ in Y-basis. Finally, the measurement of $\gamma_\mu^R$ can be done by mirroring the entire circuit. The measurement results are shown in Fig. 5 for different regimes. There we compare the data from the device with noiseless classical simulations. To make the comparison more vivid, we have added a rescaled experimental curve where we compensate the decay of the signal by the same depth-exponential factor for all observables.

Next, to evaluate $T_{1,2k}$ defined in Eq. (5) we need to probe the operator $i\gamma_1\gamma_{2k} = (-1)^k Y_1 Z_2 ... Z_{k-1} Y_k$. In order to measure the Pauli string $Y_1 Z_2 ... Z_{k-1} Y_k$, we note that

$$Y_1 Z_2 \ldots Z_{k-1} Y_k = (-1)^{k-1} G_1^\dagger \mathcal{Z}_k^L Y_k G_1 \tag{13}$$

where $G_i = H_i S_i^\dagger$, where $S_i$ is S-gate and $H_i$ the Hadamard gate applied to qubit $i$. Thus, the procedure of measuring this operator is the same as $\mathcal{Z}_k^L Y_k$ with the difference that in the latter we apply $G_1$ before the series of $U^{XY}$ gates.

Finally, applying the unitary $\exp(-\alpha\gamma_1\gamma_{2N})$, which is necessary for generating $U_{n\alpha}$ in Eq. (7), can be implemented in a similar manner. The implementing circuit consists of the series of gates $U_j^{XY}$ for $j = 2, ..., N-1$ followed by $\exp(-i\alpha Y_1 Y_2)$, then by the series of $U_j^{XY\dagger}$ in reverse order.

Results are obtained using IBM quantum hardware[53]. The experiments are performed on four different 27-qubit devices: *ibm_hanoi*, *ibm_montreal*, *ibm_mumbai*, and *ibm_toronto*. The number of qubits utilized for each experiment vary; Fig. 4 shows the chosen subsets. For example, we perform experiments represented by Fig. 2 using 10 qubits as the smallest system size that exhibits an overlap between unpaired Majorana modes which is smaller than the effect of noise. In

contrast, braiding experiments are performed on 5 qubits as the effect of noise is stronger due for deeper circuits. The depth of the circuits are chosen to be 11 and 21 cycles.

## Data availability

Source data are provided as a Source Data file. Source data are provided with this paper.

## Code availability

The code to run the experiment and access the data presented in this study is publicly available on GitHub using the link: https://github.com/IBM/observation-majorana.git.

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

## Acknowledgements

We thank Frank Pollmann and Bela Bauer for helpful discussions. We would also like to thank Sergey Bravyi, Zlatko Minev, and Sarah Sheldon for their help in preparing this publication. The research was partly supported by the IBM Research Frontiers Institute. This research was also supported in part by the National Science Foundation under Grant No. NSF PHY-1748958.

## Author contributions

The authors contributed equally to this work. O.S. and R.M. designed the experiment, developed the theoretical framework and wrote the paper. N.H. carried out the cloud-based experiment using IBM quantum hardware.

## Competing interests

The authors declare no competing interests.
