## [Peer Review File · Nature Communications]

REVIEWER COMMENTS

Reviewer #1 (Remarks to the Author):

The authors ran a quantum simulation on IBM quantum chips. They focused on the one-dimensional quantum spin model with periodic modulation, and map the system to a periodic quantum circuit models which can be simulated on IBM quantum chips for a small system size. Their model is equivalent to a Floquet version of Kitaev chain. They claimed that the existence of Majorana mode is confirmed in their quantum simulation. In addition, they also considered a non-adiabatic technique for simulating Majorana braiding in the quantum processor. I think some of the results are interesting, however, the authors have to carefully address all my concerns before a conclusive judgment.

This paper seems to be a quantum simulation of IBM quantum computer about their previous work (Phys. Rev. Lett. 125, 086804). I think the authors should also carefully discuss (in the main text) what is conceptually or theoretically novel compared to their Phys. Rev. Lett. 125, 086804. This is very helpful to the judgment of the value of the paper. The key novel part should be included carefully in the main text with more comprehensive discussions.

In my opinion, for the cases without too much conceptual novelty, a quantum simulation on a quantum cloud could be valuable only if either of the three points below can be satisfied. 1) The quantum simulation demonstrates a really hard problem, or shows some advantages compared to a classical computer at a certain level, or potentially hard problem (for example, only simulate the 16-18 qubit MBL system which can be simulated classically, but 20-24 qubit MBL systems cannot be simulated classically); 2) The quantum simulation results can be used to benchmark the quantum hardware themselves. 3) The quantum simulation demonstrates firstly a novel error mitigation scheme or a novel error correction scheme. For the current work about simulating Floquet Majorana and Majorana braiding, it seems that all three points above cannot be satisfied.

The authors mentioned, "This work could further serve as a superconducting circuit verification protocol for future topological quantum computation". If this is correct, the paper will have some impact (point 3 above). However, I didn't see how the verification of the quantum processor is extracted from this simulation. The author even didn't do a numerical classical simulation for the noisy circuit and make a comparison between the classical simulation and the IBM device simulation. On the other hand, there is no analytic study about their point of "circuit verification protocol". A similar analysis, for example, can be found in those about randomized benchmarking protocol and cross-entropy benchmarking protocol, etc.

We authors claim “We envision that this work can serves as a verification protocol for the future of solid-state based topological quantum computations [6,7]”. I don’t agree with the point. This work is about the quantum simulation of the Floquet Majorana systems using NISQ superconducting quantum computers. However, the motivation of solid-state based Majorana is to build a real device for topological quantum computation hardware. They are totally different. In my opinion, this work may be only helpful for understanding Floquet dynamics and Floquet engineering.

Reviewer #2 (Remarks to the Author):

In this work, the authors have set up a quantum simulator using noisy quantum hardware, with which a periodically driven Kitaev chain hosting both Majorana zero modes and pair modes has been effectively simulated. The authors were able to measure the wavefunctions of the Majorana zero modes through the Fourier transform of the multi-quit observables in the system. The authors further proposed to distinguish the topologically nontrivial Majorana zero modes from the trivial zero modes using two-point correlation functions, which has also been experimentally verified. In the end, the authors proposed a new scheme for the braiding of Majorana zero modes, i.e., the “Fast Approximate Swap” (FAS) method, to approximately braid the Majorana fermions localized at the two ends of the 1D chain, which has been experimentally demonstrated to effective in the non-interacting case. The authors have presented a novel and comprehensive (combining theoretical proposal and experimental measurements) study of quantum simulations and manipulations of Floquet topological Majorana zero modes on a programmable quantum simulator. The proposal of distinguishing Majorana zero modes using two-point correlation function and the new braiding method (FAS) is especially stimulating, which may provide useful guidelines for future experimental studies in the field topological quantum computations. However, I could not recommend the publication of this manuscript in Nature Communications before the authors address the following questions & comments.

1. Most of the studies have been focused on the free-fermion regime, i.e., the situation when $\lambda=0$ ($\phi=0$). The authors briefly commented that the Majorana modes are not conserved in the presence of fermion interactions. Could the authors elaborate at this point? What are the conditions that the Majorana modes could survive (up to a sufficiently long time for experimental detection) against interactions? How the Majorana modes disappear with the increase of interaction strength? Is the FAS braiding scheme still valid in the presence of the interaction term?

2. In Fig. 2(c), when the fermion interaction term is turned on ($\phi=\pi/16$), the Majorana wave functions seem to break the reflection symmetry with respect to the center of the chain, e. g., compare ψ_{2x^R} and ψ_{2x^L} in Fig. 2(c), what is the reason?

3. The authors have commented that disorder may prevent the heating effect resulted from the periodic driving. The authors may need to discuss what is the desirable disorder strength to prevent heating effects, and in the meanwhile to keep the Majorana zero modes topologically robust.

Reviewer #3 (Remarks to the Author):

The paper reports simulations of Floquet dynamics of a spin chain using an IBM quantum computer. The system has an interesting phase diagram allowing for Majorana zero modes and their close relatives in the Floquet realm.

I really enjoyed reading the manuscript. It is very well written and convincing. The theory explanations are clear. The "experimental" realisations are impressive. The zero modes and their Floquet counterparts are observed in the Fourier spectrum of the measured matrix elements. Finally, the braiding of the Majorana modes is implemented. The size of the system (~ 20 qubits) is impressive.

I enthusiastically recommend publishing the paper as is.

Response to Reviewer #1

Reviewer #1: *“The authors ran a quantum simulation on IBM quantum chips. They focused on the one-dimensional quantum spin model with periodic modulation, and map the system to a periodic quantum circuit models which can be simulated on IBM quantum chips for a small system size. Their model is equivalent to a Floquet version of Kitaev chain. They claimed that the existence of Majorana mode is confirmed in their quantum simulation. In addition, they also considered a non-adiabatic technique for simulating Majorana braiding in the quantum processor. I think some of the results are interesting, however, the authors have to carefully address all my concerns before a conclusive judgment.”*

A: We thank the referee for the assessment of our work.

Reviewer #1: *“This paper seems to be a quantum simulation of IBM quantum computer about their previous work (Phys. Rev. Lett. 125, 086804). I think the authors should also carefully discuss (in the main text) what is conceptually or theoretically novel compared to their Phys. Rev. Lett. 125, 086804. This is very helpful to the judgment of the value of the paper. The key novel part should be included carefully in the main text with more comprehensive discussions.”*

A: Thank you for the constructive suggestion. Indeed this work, in addition to running the task on an actual quantum hardware, develops and employs a distinct approach to the problem of Majorana modes. Below, we briefly summarize these differences.

Previous theoretical work (PRL) focused on the effect of Majorana modes on observables in generic interacting spin chains. By making several assumptions about these modes, we demonstrated that their presence must result in a persistent signal, such as half-frequency oscillations, which can be detected via local observables. This was surprising, as we would expect an interacting system to thermalize rapidly after a quench. However, under certain conditions, parity symmetry renders Majorana modes extremely stable with a lifetime that increases exponentially with inverse interaction strength. This work did not address a few important problems such as distinguishing a real Majorana from a trivial edge oscillating mode.

Our current manuscript solves a converse problem: by observing long-lived signals on noisy quantum hardware, we restore the Majorana modes. In this work, we directly verify that these modes are indeed topological modes (instead of, say, oscillations of a decoupled qubit) and propose a method, which we think is new, for observing their exchange that is compatible with present-day noisy devices. This advance clearly goes beyond our previous theory paper and opens new possibilities in the simulation of topological quantum computing on noisy hardware. Furthermore, we view this work as a framework for measuring local integrals of motion in genuine, non-fine-tuned quantum systems.

We have made the following changes on page 3 in response to this comment and other comments made by Reviewers:

In the interacting case $\varphi \neq 0$, Majorana mode operators are not conserved across the spectrum, i.e. $U_F^\dagger \Gamma_s^\omega U_F - e^{-i\omega} \Gamma_s^\omega = O(\tau^{-1})$. As the result, the observables associated with topological modes must decay with characteristic lifetime τ . As was shown in Ref. [32], if the bulk has zero dispersion ($\phi = 0$ or $\theta = 0$), for small interaction angles φ the lifetime diverges as $\tau \propto \mathcal{O}(\exp(c/\varphi))$, where the constant c depends on the details of interaction. In practice, the lifetime may exceed dozens of Floquet cycles even if the bulk has finite dispersion and interactions are not too strong. This approximate conservation of Majorana modes leads to the persistent signal for some local observables when the rest reach infinite-temperature values. The primary goal of this work is to use this long-lived signal to restore the structure of the modes from the experiment. In this case we look for Majorana modes of the form $\Gamma_s = \sum_{\mu=1}^{2N} \psi_\mu^s \gamma_\mu$, where ψ_μ^s are real-valued wavefunctions. We also develop a method to distinguish trivial and topological modes.

Reviewer #1: *“In my opinion, for the cases without too much conceptual novelty, a quantum simulation on*

a quantum cloud could be valuable only if either of the three points below can be satisfied. 1) The quantum simulation demonstrates a really hard problem, or shows some advantages compared to a classical computer at a certain level, or potentially hard problem (for example, only simulate the 16-18 qubit MBL system which can be simulated classically, but 20-24 qubit MBL systems cannot be simulated classically); 2) The quantum simulation results can be used to benchmark the quantum hardware themselves. 3) The quantum simulation demonstrates firstly a novel error mitigation scheme or a novel error correction scheme. For the current work about simulating Floquet Majorana and Majorana braiding, it seems that all three points above cannot be satisfied.”

A: We thank the Reviewer for this comment. In response, we note that we believe our work meets some of the criteria listed above, as well as providing sufficient conceptual novelty. Due to the fact that the interacting model we study is not integrable, it is potentially hard for classical simulations. The same experiment can be performed with > 50 qubits with feasible improvements in coherence times. To address the second criterion, we included some experimental data to the Methods section that could be used to evaluate the device’s performance. Finally, the purpose of this research is to develop a set of original tools that can be applied to generic multi-qubit quantum systems. For example, studied methods of extracting the Majorana modes can be applied to the study of local integrals of motion in many-body localized systems (see similar theoretical analysis in [52]). Also, the proposed original approximate braiding scheme does not require adiabaticity to observe particle exchange statistics.

To briefly address the third criterion, one of the main strengths of our approach is the degree to which our generated Majorana modes are robust in the presence of noise. Thus we decided to make genuine noisy devices (with no added error mitigation) a main focus of our current research.

In the Supplementary Material, we added a discussion of noise and its effects on the measured wavefunction.

We made the following changes to address this comment on page 6:

This work can be extended to regimes beyond the current classical simulation capabilities. By using devices with higher connectivity, it is possible to study generic two-dimensional materials with a broader variety of topological phases and to explore new possibilities for topological quantum computation. Further, the method studied for extracting the Majorana modes may be extended to the study of local integrals of motion in many-body localized systems, similarly to the proposal in Ref. [52].

We also added Fig. 5 to the Methods section that provides more experimental data along with the comparison to the noiseless simulations. We added changes in pages S3 and S4 of the Supplementary Material.

“The authors mentioned, “This work could further serve as a superconducting circuit verification protocol for future topological quantum computation”. If this is correct, the paper will have some impact (point 3 above). However, I didn’t see how the verification of the quantum processor is extracted from this simulation. The author even didn’t do a numerical classical simulation for the noisy circuit and make a comparison between the classical simulation and the IBM device simulation. On the other hand, there is no analytic study about their point of “circuit verification protocol”. A similar analysis, for example, can be found in those about randomized benchmarking protocol and cross-entropy benchmarking protocol, etc.”

A: We thank the Reviewer for their thorough reading and agree that this sentence should be rewritten. The term “verification” is not used here for a technical evaluation of the device’s performance. We used it in a different sense: the proposed model can be used to examine topological models. We replaced it with a more precise statement.

It is important to note that most information about IBM devices is publicly available, including information about their calibration [53].

We made the following changes to address this comment. In the abstract, we modified as:

This work could further be used to study topological models of matter using circuit-based simulations, and shows that long-sought quantum phenomena can be realized by anyone in cloud-run quantum simulations, whereby accelerating fundamental new discoveries in quantum science and technology.

In the Methods section we added the reference to the device information:

Results are obtained using IBM quantum hardware [53].

Reviewer #1: *“We authors claim “We envision that this work can serves as a verification protocol for the future of solid-state based topological quantum computations [6,7]”. I don’t agree with the point. This work is about the quantum simulation of the Floquet Majorana systems using NISQ superconducting quantum computers. However, the motivation of solid-state based Majorana is to build a real device for topological quantum computation hardware. They are totally different. In my opinion, this work may be only helpful for understanding Floquet dynamics and Floquet engineering.”*

A: We agree that our work is focused on the Floquet system. It is important to note that the Reviewer’s comment concerns the discussion section, where we consider the problem from a broader perspective. Based on this broad assessment, Floquet systems can be viewed as an approximate trotterization, where increasing driving frequency eventually corresponds to vanishing trotter steps and, therefore, to a local Hamiltonian evolution. Using this local Hamiltonian, one can model a solid-state nanowire Majorana device. This can be accomplished by considering a qubit ladder that represents two spin values (up and down) along with local gates capable of reproducing particle hopping, spin-orbital coupling, and density-density interactions. If there is sufficient coherence time, such simulations can be used to predict the performance of solid-state devices.

We made the following changes to address this comment on page 6:

Solid state systems such as nanowire devices [6, 7] can be studied through continuous-time local Hamiltonian simulations. Floquet systems similar to those studied in this work, in the limit of large frequency, are equivalent to such simulations. A potential model of a nanowire could incorporate a qubit “ladder” representing the two spin values (up and down) and local gates that account for hopping, spin-orbital coupling, and density-density interactions.

Response to Reviewer #2

Reviewer #2: *“In this work, the authors have set up a quantum simulator using noisy quantum hardware, with which a periodically driven Kitaev chain hosting both Majorana zero modes and pair modes has been effectively simulated. The authors were able to measure the wavefunctions of the Majorana zero modes through the Fourier transform of the multi-qubit observables in the system. The authors further proposed to distinguish the topologically nontrivial Majorana zero modes from the trivial zero modes using two-point correlation functions, which has also been experimentally verified. In the end, the authors proposed a new scheme for the braiding of Majorana zero modes, i.e., the “Fast Approximate Swap” (FAS) method, to approximately braid the Majorana fermions localized at the two ends of the 1D chain, which has been experimentally demonstrated to effective in the non-interacting case. The authors have presented a novel and comprehensive (combining theoretical proposal and experimental measurements) study of quantum simulations and manipulations of Floquet topological Majorana zero modes on a programmable quantum simulator. The proposal of distinguishing Majorana zero modes using two-point correlation function and the new braiding method (FAS) is especially stimulating, which may provide useful guidelines for future experimental studies in the field topological quantum computations.”*

A: We thank the Reviewer for the detailed summary of our work.

Reviewer #2: *“However, I could not recommend the publication of this manuscript in Nature Communications before the authors address the following questions & comments.*

1. Most of the studies have been focused on the free-fermion regime, i.e., the situation when $\lambda = 0$ ($\varphi = 0$). The authors briefly commented that the Majorana modes are not conserved in the presence of fermion interactions. Could the authors elaborate at this point? What are the conditions that the Majorana modes could survive (up to a sufficiently long time for experimental detection) against interactions? How the Majorana modes disappear with the increase of interaction strength? Is the FAS braiding scheme still valid in the presence of the interaction term?"

A: We would like to thank the referee for their thorough questions. In our previous theoretical work [32], we addressed most of these questions. We summarize them here and have made changes to the manuscript.

- “The authors briefly commented that the Majorana modes are not conserved in the presence of fermion interactions. Could the authors elaborate at this point?” In free-fermion models, Majorana modes are stable, their operators $\Gamma_\alpha = \sum_\mu \psi_\mu^\alpha \gamma_\mu$ expressed through the eigenstates ψ_μ^α of a single-fermion Hamiltonian. However, topological order is believed to be not stable across the spectrum, i.e. in general $U_F \Gamma_\alpha \neq \pm \Gamma_\alpha U_F$. This leads to decay of the expectation of Majorana modes into collective bulk modes, i.e. $U_F^{\dagger 2n} \Gamma_\alpha U_F^{2n} = e^{-2n/\tau} \Gamma_\alpha + \dots$, where τ is the lifetime.
- “What are the conditions that the Majorana modes could survive (up to a sufficiently long time for experimental detection) against interactions? How the Majorana modes disappear with the increase of interaction strength?” In our previous work, we considered the case when the bulk has zero dispersion (essentially if $\phi = \pi/2$ or $\theta = \pi/2$) and Majorana modes are well separated (i.e. have zero overlap). For small λ we had shown that the lifetime scales as

$$\tau = \mathcal{O}(\exp(c/\varphi)) \quad (1)$$

where $\varphi = \lambda T$, T is the driving period, and c is the constant that depends on the spectral norm of the interactions and its locality. This is, however, the asymptotic result valid for small λ . In practice, setting even $\varphi \equiv \lambda T = \pi/16$ provides the lifetime $\gg 10$ Floquet circuit cycles.

- “Is the FAS braiding scheme still valid in the presence of the interaction term?” Yes. The proof of Eq. (8) does not rely on the free-fermion nature of the evolution. However, the observation times must be much smaller than the lifetime τ .

In response to this comment and other Reviewer comments, we added the following paragraph on page 3:

In the interacting case $\varphi \neq 0$, Majorana mode operators are not conserved across the spectrum, i.e. $U_F^\dagger \Gamma_s^\omega U_F - e^{-i\omega} \Gamma_s^\omega = \mathcal{O}(\tau^{-1})$. As the result, the observables associated with topological modes must decay with characteristic lifetime τ . As was shown in Ref. [32], if the bulk has zero dispersion ($\phi = 0$ or $\theta = 0$), for small interaction angles φ the lifetime diverges as $\tau \propto \mathcal{O}(\exp(c/\varphi))$, where the constant c depends on the details of interaction. In practice, the lifetime may exceed dozens of Floquet cycles even if the bulk has finite dispersion and interactions are not too strong. This approximate conservation of Majorana modes leads to the persistent signal for some local observables when the rest reach infinite-temperature values. The primary goal of this work is to use this long-lived signal to restore the structure of the modes from the experiment. In this case we look for Majorana modes of the form $\Gamma_s = \sum_{\mu=1}^{2N} \psi_\mu^s \gamma_\mu$, where ψ_μ^s are real-valued wavefunctions. We also develop a method to distinguish trivial and topological modes.

We also added a clarification to page 5:

This method applies in both the interacting and non-interacting regimes.

Reviewer #2: “2. In Fig. 2(c), when the fermion interaction term is turned on ($\varphi = \pi/16$), the Majorana wave functions seem to break the reflection symmetry with respect to the center of the chain, e. g., compare ψ_{2x}^R and ψ_{2x}^L in Fig. 2(c), what is the reason?”

A: To be precise, the theoretical curves (solid lines) are completely symmetric. The comments seem to concern the experimental data points. The source of asymmetry is that the experimental system is not symmetric itself due to coherent errors and fluctuations caused by the noise. For example, Ref. [53] provides the details of the devices for latest calibration, which show that qubits exhibit different level of noise. The visibly higher level of noise for interacting experiment can be explained by deeper circuits due to the additional layer of ZZ-gates. As such, the fluctuations leading to asymmetry astutely observed by the Reviewer is quite normal, and in fact expected in the presence of noise.

We added the following sentence to clarify this observation to page 4:

In the interacting regime, where we add an extra set of noisy two-qubit ZZ gates in each Floquet cycle, we expect to see a visibly higher level of noise in the resulting wavefunction as can be seen in Fig. 2c. More data assessing the device's performance is presented in the Methods section.

We also added Fig. 5 that demonstrates raw data. This figure shows the increased effect of noise in interacting regime.

Reviewer #2: “3. The authors have commented that disorder may prevent the heating effect resulted from the periodic driving. The authors may need to discuss what is the desirable disorder strength to prevent heating effects, and in the meanwhile to keep the Majorana zero modes topologically robust.”

A: Adding disorder such as randomization of phases in Z gates, i.e. $\phi \rightarrow \phi + \delta_i$, where $\delta_i \in [-W, W]$, can reduce the heating due to the (finite-size) many-body localization (MBL) phenomenon. However, such a simplistic scheme may require strong disorders $W/\theta \sim 10$ that can cause transition into trivial phase. Designing such systems requires finding good model parameters such as in [51]. Additionally, one could explore more sophisticated schemes as adding binary X-field disorder that do not violate parity symmetry but reduces heating [32].

Disorder may cause additional problems: it creates localized states overlapping in frequency with the Majorana modes. Measuring Majorana modes in this case remains an interesting open problem.

To reflect this comment, we made the following changes on page 5:

The finite lifetime of the Majorana modes is attributed to natural tendency of Floquet systems to “heat up”. Adding disorder such as randomization of phases in Z gates, i.e. $\phi \rightarrow \phi + \delta_i$, where $\delta_i \in [-W, W]$, can reduce the heating because of the many-body localization (MBL) phenomenon [50]. However, such a simplistic scheme may require disorder values W that can cause transition into the trivial phase. Avoiding phase transition would require finding good model parameters [51] or using more sophisticated techniques [32].

Response to Reviewer #3

Reviewer #3: “The paper reports simulations of Floquet dynamics of a spin chain using an IBM quantum computer. The system has an interesting phase diagram allowing for Majorana zero modes and their close relatives in the Floquet realm.

I really enjoyed reading the manuscript. It is very well written and convincing. The theory explanations are clear. The "experimental" realisations are impressive. The zero modes and their Floquet counterparts are observed in the Fourier spectrum of the measured matrix elements. Finally, the braiding of the Majorana modes is implemented. The size of the system (20 qubits) is impressive.

I enthusiastically recommend publishing the paper as is.”

A: Thank you very much! We thank the Reviewer for their positive assessment of our work.

REVIEWERS' COMMENTS

Reviewer #1 (Remarks to the Author):

The revised version and reply resolve the major concerns shown in my last report. I believe the paper can meet the high standard of nature communications, and therefore recommend the publication of the paper.

Regarding the reply "To briefly address the third criterion, one of the main strengths of our approach is the degree to which our

generated Majorana modes are robust in the presence of noise. Thus we decided to make genuine noisy devices (with no added error mitigation) a main focus of our current research.", in fact, as long as you make something "Majorana", there are clearly certain robust things, which is not surprising and not quit *useful*. Unless this type of Majorana can create a more robust device for information storage and manipulations, which I doubt for this system. Nevertheless, the paper is still very nice.

Reviewer #2 (Remarks to the Author):

The explanations from the authors help with the appreciation of their work. Although there are still difficulties in scaling up the size of the simulation, like the finite coherent time and connectivity in superconducting quantum system, I do think this work presents a nice work in simulating Majorana modes in the presence of interaction and noise as well as realizing braiding of these modes. Now, I would definitely recommend publishing the paper on Nature Communication and would expect to see it inspires more work in pushing forward our understanding of Floquet system and even topological quantum computation.